# Gamma-Tocopherol: A Comprehensive Review of Its Antioxidant, Anti-Inflammatory, and Anticancer Properties

**DOI:** 10.3390/molecules30030653

**Published:** 2025-02-01

**Authors:** Basma Es-Sai, Hicham Wahnou, Salma Benayad, Soufiane Rabbaa, Yassir Laaziouez, Riad El Kebbaj, Youness Limami, Raphaël Emmanuel Duval

**Affiliations:** 1Sciences and Engineering of Biomedicals, Biophysics and Health Laboratory, Higher Institute of Health Sciences, Hassan First University, Settat 26000, Morocco; essaibassma1999pfe@gmail.com (B.E.-S.); salmabenayad6@gmail.com (S.B.); rabbaasoufian@gmail.com (S.R.); y.laaziouez@gmail.com (Y.L.); elkebbajriad@gmail.com (R.E.K.); youness.limami@uhp.ac.ma (Y.L.); 2Laboratory of Immunology and Biodiversity, Faculty of Sciences Ain Chock, Hassan II University, B.P. 2693, Maarif, Casablanca 20100, Morocco; hwwahnou@gmail.com; 3Université de Lorraine, F-54000 Nancy, France

**Keywords:** γ-tocopherol, antioxidant, anti-inflammatory, anticancer, clinical trials, nanoformulations

## Abstract

Gamma-tocopherol (γ-tocopherol), a major isoform of vitamin E, exhibits potent antioxidant, anti-inflammatory, and anticancer properties, making it a promising therapeutic candidate for treating oxidative stress-related diseases. Unlike other tocopherol isoforms, γ-tocopherol effectively neutralizes reactive oxygen species (ROS) and reactive nitrogen species (RNS), providing robust cellular protection against oxidative damage and lipid peroxidation. Its anti-inflammatory effects are mediated through the modulation of pathways involving cyclooxygenase-2 (COX-2) and tumor necrosis factor-alpha (TNF-α), reducing chronic inflammation and its associated risks. In cancer therapy, γ-tocopherol demonstrates multifaceted activity, including the inhibition of tumor growth, induction of apoptosis, and suppression of angiogenesis, with significant efficacy observed in cancers such as prostate, lung, and colon. Preclinical and clinical studies support its efficacy in mitigating oxidative stress, inflammation, and cancer progression, with excellent tolerance at physiological levels. However, high doses necessitate careful evaluation to minimize adverse effects. This review consolidates current knowledge on γ-tocopherol’s biological activities and clinical implications, underscoring its importance as a natural compound for managing inflammation, oxidative stress, and cancer. As a perspective, advancements in nanoformulation technology could enhance γ-tocopherol’s bioavailability, stability, and targeted delivery, offering the potential to optimize its therapeutic application in the future.

## 1. Introduction

Gamma-tocopherol (γ-tocopherol) is one of the eight naturally occurring isoforms of vitamin E, a fat-soluble compound recognized for its essential role in maintaining cellular health and combating oxidative damage [1]. Unlike alpha-tocopherol (α-tocopherol), the most studied and abundant form of vitamin E, γ-tocopherol possesses distinct chemical and biological properties that make it uniquely effective in certain physiological contexts [1]. Found abundantly in dietary sources such as nuts, seeds, and vegetable oils (Figure 1A), γ-tocopherol is distinguished by its ability to neutralize reactive nitrogen species, particularly peroxynitrite, a feature less prominent in other tocopherol isoforms [1,2].

Oxidative stress, characterized by an imbalance between reactive oxygen and nitrogen species (ROS and RNS) and the body’s antioxidant defenses, is a key driver of cellular damage and dysfunction [3]. This pathological state contributes to the progression of numerous chronic diseases, including cardiovascular disorders, neurodegenerative conditions, and cancer [4,5]. Closely linked to oxidative stress, chronic inflammation acts as a mediator in many of these diseases. Persistent inflammatory signals promote tissue damage and creates a microenvironment conducive to the development and progression of disorders such as arthritis, atherosclerosis, and malignancies [6]. In cancer, oxidative stress and inflammation synergistically drive tumorigenesis by promoting DNA damage, genetic instability, and immune evasion [7]. Despite advancements in conventional therapies, such as nonsteroidal anti-inflammatory drugs (NSAIDs) and chemotherapeutic agents, these treatments are often associated with significant adverse effects [8]. Long-term use of NSAIDs, for instance, is linked to gastrointestinal and cardiovascular risks, while many anticancer agents cause severe systemic toxicity, leading to compromised quality of life and treatment discontinuation [8,9,10]. These limitations underscore the pressing need for safer, more targeted therapeutic strategies that mitigate disease progression without imposing undue harm on patients.

γ-tocopherol has emerged as a promising candidate for addressing these challenges due to its multifaceted biological properties [2]. Its potent antioxidant activity not only neutralizes harmful ROS and RNS but also interrupts the vicious cycle of oxidative stress and inflammation [11]. Furthermore, its ability to modulate inflammatory pathways and disrupt cancer cell proliferation positions it as a natural compound with therapeutic potential across a spectrum of oxidative stress and inflammation-driven diseases [12]. Clinical studies have begun to reveal its promise in reducing disease progression and improving patient outcomes in various pathological contexts [13]. The emergence of nanoformulation technologies further enhances the therapeutic potential of γ-tocopherol. By improving its bioavailability, stability, and targeted delivery, nanoformulations offer a means to maximize its efficacy while minimizing potential side effects [14]. This review examines the broad therapeutic spectrum of γ-tocopherol, consolidates evidence from clinical studies, and explores the innovative role of nanoformulations in advancing its application as a modern therapeutic agent.

## 2. Structure and Properties

Tocopherols are a group of lipid-soluble compounds classified under vitamin E [15]. They are recognized for their potent antioxidant properties, which play a crucial role in protecting cell membranes from the oxidative damage caused by free radicals. Structurally, tocopherols are characterized by a chromanol ring attached to a long hydrophobic phytyl tail, making them amphipathic molecules [16]. This allows tocopherols to embed within biological membranes, where they stabilize lipid bilayers and prevent lipid peroxidation. There are four main types of tocopherols [17], alpha (α), beta (β), gamma (γ), and delta (δ), which differ in terms of the number and position of methyl groups on the chromanol ring (Figure 1B).

α-tocopherol: the chromanol ring has methyl groups at positions R1 = CH₃ and R2 = CH₃. This fully methylated structure allows α-tocopherol to have the highest antioxidant activity among tocopherols and makes it the most bioavailable form.β-tocopherol: structurally, β-tocopherol has one methyl group located at R1 = CH₃ and one hydrogen atom in R2 = H. Its activity is intermediate, between that of α-tocopherol and γ-tocopherol, due to this arrangement.γ-tocopherol: in this variant, the hydrogen atom is located at R1 = H, while the methyl group is located at R2 = CH₃. The absence of a methyl group at R1 alters its antioxidant properties. While its radical-scavenging activity is slightly lower than α-tocopherol, γ-tocopherol has a unique ability to trap RNS, such as peroxynitrite, which may confer additional protective effects.δ-tocopherol: the chromanol ring in δ-tocopherol has hydrogen atoms in both substituents, which are positioned at R1 = H and R2 = H. This structure results in δ-tocopherol having the highest ability to neutralize ROS among the tocopherol isomers.

γ-tocopherol is distinguished by its unique chemical structure and physical properties, which together underpin its essential biological roles [1]. Its structure features a chromanol ring as the aromatic head, containing a hydroxyl group (-OH) at the 6th position (Figure 1B). This hydroxyl group is critical for its antioxidant activity, as it donates hydrogen atoms to neutralize free radicals and prevent oxidative damage [18]. γ-tocopherol’s substituent pattern, with methyl groups at the 7th and 8th positions, differs from α-tocopherol, which also has a methyl group at the 5th position (Figure 1B). This structural variation influences γ-tocopherol’s antioxidant function and biological activity. Additionally, the molecule possesses a long hydrophobic phytyl tail attached to the chromanol ring [19]. This tail anchors the molecule within lipid bilayers of cell membranes, where it protects membrane lipids from peroxidation and maintains cellular integrity [19]. Its molecular formula is C_28_H_48_O_2_, with a molecular weight of 416.68 g/mol. γ-tocopherol is lipophilic, making it soluble in fats and oils but insoluble in water [20].

Physically, γ-tocopherol appears as a viscous oil with a yellow to amber color. It remains stable under normal physiological conditions but can decompose at high temperatures. While it is insoluble in water, it dissolves readily in organic solvents such as ethanol, acetone, and lipids, facilitating its use in lipid-based applications [11]. γ-tocopherol is relatively stable when exposed to heat and light but is prone to degradation upon prolonged exposure to ultraviolet radiation or oxygen [21,22]. This susceptibility can diminish its antioxidant efficacy over time.

## 3. Antioxidant Effects

Oxidative stress reflects an imbalance between the systemic manifestation of the reactivity of oxygen species and the ability of biological systems to detoxify the body from free radicals, which leads to the accumulation of ROS in cells, resulting cellular toxicity and subsequently leading to tissue damage [3]. Antioxidants play a crucial role in neutralizing ROS and, among them, γ-tocopherol stands out due to its potent radical-scavenging capabilities [23,24]. Its antioxidant activity is attributed to its unique chemical structure, particularly the chromanol ring, which facilitates the donation of hydrogen atoms and stabilization of the resulting radical species [23,24].

Density functional theory (DFT) has been employed to explore the thermodynamic parameters governing the antioxidant activity of γ-tocopherol [25], suggesting that γ-tocopherol possesses a lower band dissociation than α-tocopherol [26]. This property may enhance the efficiency of γ-tocopherol in donating a hydrogen atom from its phenolic hydroxyl group, which can facilitate free radical neutralization. In addition, the electronic configuration of γ-tocopherol characterized by one less methyl group compared to α-tocopherol, enables it to trap more effectively with RNS, such as peroxynitrite (ONOO-) [27]. Furthermore, key parameters include the bond dissociation enthalpy (BDE), which measures the ease of hydrogen donation, with a low BDE indicating high reactivity towards free radicals [25,28]. The ionization potential (IP) reflects the compound’s electron-donating capacity, with lower IP values enhancing radical scavenging via electron transfer [29].

Theoretical studies also highlight two dominant mechanisms of action: (i) hydrogen atom transfer (HAT), where the phenolic hydrogen atom neutralizes free radicals [30], and (ii) sequential proton loss electron transfer (SPLET), which involves deprotonation followed by electron transfer [31]. These mechanisms underscore γ-tocopherol’s versatility and effectiveness in mitigating oxidative stress and preventing cellular damage.

Various studies have demonstrated, experimentally, the protective effects of γ-tocopherol against oxidative stress through its impact on key markers of this biological process involved in the apparition of several pathologies (Table 1).

### 3.1. Effects Against Lipid Peroxidation

ROS cause lipid peroxidation, which is a complex process, renowned for the rearrangement of doubles bonds in polyunsaturated fatty acids, particularly arachidonic acids. This reaction generates an inactivation of receptors, and membrane permeability and fluidity, which leads to apoptosis [44,45].

γ-tocopherol exhibits remarkable antioxidant properties through its ability to modulate lipid peroxidation markers, particularly malondialdehyde (MDA) levels, in diverse biological systems. In diabetic-afflicted mice kidneys, γ-tocopherol was shown to decrease MDA concentrations [32]. Similarly, in normal mice stimulated by lipopolysaccharide (LPS), γ-tocopherol attenuated lipid peroxidation in the liver [33], highlighting its preventive and protective effect across different tissues. In sheep exposed to burn and smoke inhalation, γ-tocopherol nebulization decreased MDA levels in the lungs [34], reflecting its impact in modulating lipid peroxidation even under environmental toxin exposure. Interestingly, when γ-tocopherol is combined with α-tocopherol, MDA levels decreased in the plasma and livers of chicken, suggesting that this interaction might influence specific antioxidant pathways [46]. In light of this, natural sources such as argan oil, abundant in antioxidant compounds like γ-tocopherol [47], are recognized for their restorative properties in cases of oxidative stress [48,49]. Previous studies demonstrate that argan oil can normalize depleted MDA levels in the livers and the brains of mice under oxidative stress challenges, reaffirming the role of diet-derived antioxidants in hepatoprotection [49]. In addition, it was demonstrated that γ-tocopherol attenuated the serum levels of 8-isoprostane in rats implanted with 17β-estradiol (E2), with impacts varying based on the concentration and the period of treatment [33,35] (Figure 2 and Figure 3). Clinically, γ-tocopherol has been explored in conditions like metabolic syndrome, where it regulates lipid peroxidation by decreasing plasmatic MDA+HNE (4-hydroxy 2-nonenal) and lipid peroxides levels [36]. Given the relationship between γ-tocopherol and lipid peroxidation, it has been well-explained how this component can limit this process. One key mechanism involves the formation of metabolites from its oxidation, such as tocopherylquinone, which can scavenge lipid peroxyl radicals (ROO^.^) and stop lipid peroxidation [2].

This suggests that γ-tocopherol successfully helps to attenuate oxidative stress by regulating lipid peroxidation through reducing and normalizing the markers involved in this process, offering a potential beneficial therapeutic for oxidative stress-related disorders.

### 3.2. Effects Against DNA Oxidative Damage

ROS, particularly hydroxyl radical (OH•), has the potential to induce DNA damage, which encompasses processes such as base mutation, single- and double-strand breaks, and the formation of DNA adducts [37].

γ-tocopherol has been evaluated in preclinical models, demonstrating its effectiveness in protecting against DNA damage induced by oxidative stress. For instance, in the mammary gland of rats, γ-tocopherol administration significantly attenuated the E2-induced 8-oxo-2′-deoxyguanosine (8-oxo-dG) level, which is a marker of DNA oxidative damage and a product of nitrogenous base oxidation [37], by 72% and 67% at 7 and 14 days of treatments, respectively [35]. Similarly, Chen et al. [38] found that γ-tocopherol lowered the 8-oxo-dG level by 35% in colon tumors and by 42% in the adjacent tissues. On the other hand, the level of 8-oxo-dG was markedly decreased in the wild-type and nuclear factor erythroid 2–related factor 2 (Nrf2) (−/−) epithelial cells of mice, which suggests that the antioxidant potential of γ-tocopherol is independent of Nrf2-regulated antioxidative enzymes [39] (Figure 2). Additionally, it was demonstrated that the supplementation of γ-tocopherol significantly attenuated the levels of the marker of DNA damage phosphorylated histone γH2AX in the colon of 2-amino-1-methyl-6-phenylimidazo(4,5-b) pyridine (PHIP) -treated mice [38]. According to these available studies, γ-tocopherol exerts a protective effect against DNA oxidative damage, which is primarily demonstrated in vivo. These studies suggest an indirect effect of γ-tocopherol is to preserve DNA integrity and stability. However, the direct interactions between γ-tocopherol and DNA have not been conclusively demonstrated. Its role as an antioxidant highlights its preventive effect in maintaining DNA structure. Further research is needed to elucidate the direct interaction between γ-tocopherol and DNA to gain a deeper understanding of the associated biological pathways (Figure 2).

### 3.3. Effect on Antioxidant Defense

The body has a complex system of antioxidant defenses to protect against the harmful effects of ROS and RNS. There are two distinct levels of antioxidant effects: one is provided by nutrition through vegetable and fruits rich in vitamins C, E, carotenoids, flavonoids, or glutathione, whose role is to scavenging free radicals; the other is endogenous and consists of antioxidant enzymes such as superoxide dismutase (SOD), catalase (Cat), glutathione peroxidase (GPx), and heme oxygenase (HO-1), whose role is to degrade free radicals [50,51,52].

γ-tocopherol treatment is quite successful in improving the oxidative state in the kidneys of diabetic mice by normalizing the protein expression of antioxidant enzymes (catalase, GPx, and HO-1) [32]. At the transcriptional level, Das Gupta et al. found that the supplementation of γ-tocopherol significantly upregulating the mRNA expression of antioxidant enzymes (SOD1, GPx, and catalase) in the mammary glands of rats treated with E2, which can lead to degradation of the ROS produced by E2 [35]. Moreover, an in vitro study revealed that γ-tocopherol neutralizes ROS levels in PBMCs stimulated by LPS [40]. Similarly, the arterial superoxide anion generation, a well-established substrate of SOD enzyme, was significantly decreased in the γ-tocopherol-fed rats [53], suggesting that its degradation is facilitated by the enhancement of this enzyme. In addition, it was demonstrated that there was a notable rise in SOD activity in aortic homogenates as well as in the plasma of rats administered by γ-tocopherol [53]. Furthermore, γ-tocopherol significantly increases the proteins levels of NQO1 (NAD(P)H quinone dehydrogenase 1), GCLM (glutamate-cysteine ligase modifier subunit), and HO-1 in mammary tumors in mice subjected to E2-induced alterations, thereby mitigating oxidative stress, which helps to prevents cellular homeostasis [35] (Figure 2).

### 3.4. Effect on Nrf2 Pathway

Nrf2 is a transcription factor that is key for the preservation of cellular redox homeostasis and the elimination of reactive and carcinogenic mediators. The connection between Nrf2 and oxidative stress has been thoroughly elucidated by El Kebbaj et al. [54], who highlighted how natural compounds like fatty acids, tocopherols (α, β, γ, and δ), phytosterols (particularly schottenol and spinasterol), and ferulic acid can regulate Nrf2 expression during the oxidative stress process.

In this regard, it was demonstrated that γ-tocopherol-enhanced mammary gland Nrf2 mRNA levels in rats treated with E2 are correlated by increasing the mRNA expression of antioxidant enzyme [35]. Furthermore, a diet enriched with γ-tocopherol mixtures was shown to increase Nrf2 at both the mRNA and protein levels in the livers of mice [41,42]. However, it was also evaluated that Nrf2 did not exhibit a direct correlation with the modulation of MDA levels in the livers of mice [42], suggesting that the mechanism governing these pathways may be independent or may involve additional mediators indirectly linked to Nrf2 modulation. To better understand how γ-tocopherol influences Nrf2, a mechanistic study revealed that a tocopherol-rich mixture containing γ-tocopherol maintains the mRNA and protein levels of Nrf2 by inhibiting CpG hypermethylation in the Nrf2 promoter during prostate carcinogenesis in mice and in TRAMP-C1 cells [43] (Figure 2). These studies suggest that γ-tocopherol inhibits oxidative stress through modulating the Nrf2 signaling pathway to upregulate antioxidant proteins involved in free radical neutralization.

## 4. Anti-Inflammatory Activity

Inflammation is a key immune response that helps repair tissues after injury, but chronic inflammation can increase the risk of various diseases. γ-tocopherol, a powerful antioxidant, has shown potential in reducing inflammation by limiting neutrophils infiltration and cytokines’ production [55,56]. Several studies have evaluated its effects, revealing that γ-tocopherol may help modulate inflammatory responses, offering therapeutic benefits in conditions associated with acute inflammation and also with chronic inflammation such as diabetes and asthma.

### 4.1. Carrageenan-Induced Inflammation

γ-tocopherol’s effects in acute inflammation models, where the inflammatory response is more immediate and transient, have been thoroughly evaluated. In the carrageenan-induced inflammation model in the air pouch, γ-tocopherol significantly reduced the pro-inflammatory eicosanoids prostaglandin E2 (PGE2) (46%) and Leukotriene B4 (LTB4) (70%), the latter being a potent chemotactic factor produced by 5-lipoxygenase in neutrophils [57]. At a higher dose (i.e., 100 mg/kg), γ-tocopherol exhibited similar inhibitory effects on PGE2 (51%) and LTB4. Additionally, γ-tocopherol’s effects on TNF-α, a key inflammation mediator, were investigated. At the lower dose (i.e., 33 mg/kg), γ-tocopherol did not significantly inhibit TNF-α (51%), while at the higher dose, γ-tocopherol reduced TNF-α by 65%. These results suggest that γ-tocopherol modulates multiple components of the inflammatory response, offering potential therapeutic benefits in acute inflammation settings (Figure 3).

In a carrageenan-induced inflammation model in rats, a combination of aspirin and γ-tocopherol exhibited enhanced anti-inflammatory effects compared to aspirin alone [58]. The combination reduced PGE2 levels by 40% and decreased exudate volume by 15%, suggesting a prolonged anti-inflammatory action [58]. However, no significant effects were observed on immune cell infiltration. Of note, the combination of aspirin and γ-tocopherol also partially alleviated aspirin-induced gastric lesions and spared the reduction in food intake typically caused by aspirin. Additionally, aspirin and γ-tocopherol reduced gastric PGE2 depletion and decreased 8-isoprostane levels, indicating a protective effect against oxidative stress in the stomach [58]. These findings suggest that γ-tocopherol may improve the therapeutic profile of aspirin by enhancing its anti-inflammatory efficacy and reducing its adverse gastrointestinal effects.

In the IFN-γ/PMA-stimulated Caco-2 intestinal cell model system, γ-tocopherol exhibits significant anti-inflammatory effects, particularly by reducing IL-8 at both mRNA and protein levels [59]. In fact, at concentrations of 10 and 100 μM, γ-tocopherol significantly decreases IFN-γ/PMA-induced IL-8 production without inducing cytotoxicity. Additionally, γ-tocopherol downregulates IL-8 mRNA expression at 8 h, although this effect diminishes by 24 h [59]. Regarding NF-κB activation, γ-tocopherol is an inhibitor with moderate effectiveness, and δ-tocopherol the least effective [59]. Globally, these results highlight γ-tocopherol as a potent anti-inflammatory tocopherol for modulating inflammatory signaling pathways in this model.

In murine RAW264.7 macrophages treated with LPS and human A549 epithelial cancer cells treated with IL-1β, γ-tocopherol and its metabolite gamma-carboxyethyl hydroxychroman (γ-CEHC) effectively inhibit PGE2 synthesis in a dose-dependent manner, with γ-tocopherol demonstrating greater potency (IC_50_ values: 7.5 ± 2 μM in macrophages and 4 ± 1 μM in A549 cells) [60]. γ-tocopherol also reduces prostaglandin D2 (PGD2) levels in LPS-stimulated macrophages but is slightly less effective than it is on PGE2. While α-tocopherol has minimal or no effect on PGE2 and PGD2 synthesis, γ-CEHC directly inhibits COX-2 activity after short exposures, while γ-tocopherol requires prolonged exposure [60]. However, none of the tocopherols significantly reduce COX-2 protein expression or mRNA levels, despite inflammatory stimulation (Figure 3).

TNF-α stimulation increased Vascular Cell Adhesion Molecule 1 (VCAM-1) expression in HMEC-1 cells, which was significantly reduced by all tocopherols at 20 and 40 μM, with γ-tocopherol showing the strongest effect. At 10 μM, none of the tocopherols affected VCAM-1 expression. Moreover, in lymphatic endothelial cells, tocopherols did not reduce VCAM-1 expression after TNF-α exposure [61] (Figure 4).

γ-tocopherol exhibits potent anti-inflammatory effects by modulating key inflammatory markers such as IL-8 and VCAM-1, with significant inhibition of NF-κB activation and COX-2 activity. The mechanism of action of γ-tocopherol seems to involve both direct inhibition of inflammatory mediators and modulation of the endothelial barrier function, highlighting its therapeutic potential in inflammatory diseases and angiogenesis-related conditions [62]. Its effects were more pronounced than those of α-tocopherol, making γ-tocopherol a promising candidate for further exploration in inflammatory and vascular health management.

### 4.2. Diabetes-Induced Inflammation

γ-tocopherol demonstrates significant anti-inflammatory effects in diabetic models.

The studies collectively demonstrate that γ-tocopherol exerts anti-inflammatory effects by targeting key molecular pathways, particularly in severe diabetic conditions. γ-tocopherol consistently reduced NF-κB activity across tissues, leading to decreased levels of pro-inflammatory cytokines like IL-1β, TNF-α, and Monocyte Chemoattractant Protein-1 (MCP-1) [32,63]. It also elevated Sirtuin 1 (SIRT1), which inhibits NF-κB signaling and enhances antioxidant and repair mechanisms. The reduction in inflammasome components, such as NOD-like Receptor Pyrin Domain Containing 3 (NLRP3) and caspase-1 in hepatic tissue, further supports γ-tocopherol’s role in mitigating severe inflammation. Additionally, γ-tocopherol’s ability to increase nuclear peroxisome proliferator-activated receptor gamma coactivator 1 alpha (PGC-1α) and p53 in wound healing suggests it improves mitochondrial function and cellular stress responses [63].

### 4.3. Asthmatic Disease

Asthma is one of the most prevalent chronic childhood illnesses, characterized as a heterogeneous and multifactorial disorder influenced by a complex interplay of genetic and environmental factors [64]. Many in vivo and clinical trials have been conducted to assess the potential of γ-tocopherol in improving this disease.

#### 4.3.1. Non-Allergic Asthma Induced by LPS

A combined animal and clinical study evaluated γ-tocopherol’s effects on LPS-induced airway inflammation [65]. In rats, LPS challenge-induced significant neutrophils accumulation in bronchoalveolar lavage fluid (BALF) and airway tissues, which were mitigated by 4 days of γ-tocopherol supplementation (30 mg/kg/day) [65]. A clinical trial using a double-blind, placebo-controlled crossover design involved 13 non-asthmatic individuals receiving γ-tocopherol supplementation (830 mg/day) or placebo before an LPS challenge. γ-tocopherol significantly reduced neutrophils and eosinophils in sputum and prevented LPS-induced increases in IL-1β levels, although it did not significantly alter IL-6, TNF-α, or markers such as TLR4 and CD11b [65]. These findings underline γ-tocopherol’s potential to reduce neutrophilic and eosinophilic inflammation in both animal and human models (Figure 3). Furthermore, a randomized study examined γ-tocopherol supplementation (1200 mg/day for 14 days) vs. placebo in participants undergoing an LPS challenge [66]. γ-tocopherol significantly reduced post-treatment sputum eosinophils, total mucins, and MUC5AC content when compared to the placebo group. It also attenuated LPS-induced neutrophilia at 6 h and 24 h post-challenge. While mucociliary clearance (MCC) slowed significantly during placebo treatment, no such slowing was observed during γ-tocopherol treatment [66]. Furthermore, γ-tocopherol mitigated LPS-induced increases in IL-1β and IL-8 but did not affect IL-6 or other TH1 cytokines [66]. These results underscore γ-tocopherol’s ability to reduce airway inflammation, neutrophils infiltration, and mucin production while potentially preserving MCC during LPS-induced airway challenges (Figure 3). Additionally, in a phase I open-label study, 16 volunteers (8 asthmatics and 8 healthy controls) were treated with a γ-tocopherol-rich supplement: one gel tab per day for one week, followed by two gel tabs per day for another week, with a washout period between dosing cycles. Blood samples collected before and after each dosing period showed no significant changes in serum cytokines (e.g., IL-1α, IL-1β, IL-12p40, granulocyte-macrophage colony-stimulating factor (GM-CSF), TNF, eotaxin, Rantes, macrophage inflammatory protein-1 alpha (MIP1α), and interferon gamma-inducible protein 10 (IP-10)) [40]. PBMCs collected from 14 participants and exposed to LPS exhibited significantly reduced secretion of pro-inflammatory cytokines (IL-1β, IL-6, and TNF-α) and chemokines (MCP1 and MIP1α) after γ-tocopherol supplementation, without affecting anti-inflammatory cytokines (Interleukin-1 receptor antagonist (IL-1RA) and Interleukin-10 (IL-10)) [40]. No differences in cytokine responses were observed between asthmatic and non-asthmatic participants.

#### 4.3.2. Other Asthma Inducers

In a pilot randomized clinical trial [67], volunteers aged 18–45 underwent a controlled 2 h exposure to 500 mg/m^3^ of wood smoke particulates (WSPs) to assess γ-tocopherol supplementation’s effects on airway inflammation. Treatment included four doses of 1253 mg over 2 days or daily doses of 1154–1253 mg over 7 days. WSP exposure increased sputum neutrophils at 6 and 24 h, with no significant differences between γ-tocopherol and placebo groups [67]. Unlike its effects on endotoxin-induced inflammation, γ-tocopherol did not mitigate WSP or O_3_-induced neutrophilia, suggesting it may be more effective against endotoxin-induced airway responses [67].

### 4.4. Nanoformulation

Nanoformulation strategies have emerged as a promising solution to enhance γ-tocopherol pharmacokinetics and therapeutic efficacy [68]. By incorporating γ-tocopherol into nanoparticles, liposomes, or micelles, its stability, solubility, and targeted delivery to specific tissues are significantly improved [69]. Although studies on the biological effects of γ-tocopherol nanoformulations remain limited, existing research highlights their significant potential. For instance, a study by Kuo et al. demonstrated that γ-tocopherol nanoemulsions enhance anti-inflammatory efficacy by facilitating targeted delivery to inflamed tissues. This targeted approach allows for more efficient suppression of pro-inflammatory cytokines, such as TNF-α and IL-1β. In preclinical models of inflammation induced by 2% croton oil, γ-tocopherol nanoemulsions showed a marked reduction in inflammatory markers, underscoring their therapeutic promise for managing inflammatory conditions [14].

## 5. Anticancer Activity

The anti-tumoral potential of γ-tocopherol has been extensively studied through a variety of in vitro and in vivo experimental models. The underlying mechanisms by which the molecule exerts its anticancer effects are multifaceted and involve modulation of key signaling pathways and molecular targets that are crucial for tumorigenesis and cell proliferation. In this section, we will discuss the detailed mechanistic role of γ-tocopherol in cancer chemoprevention and destruction.

### 5.1. Modulation of Cell Death

The discovery of novel cell death pathways has significantly expanded our understanding of how cells respond to various forms of cellular damage. Apoptosis and ferroptosis are two critical forms of cell death that are intricately linked in cancer treatment [70,71]. Apoptosis, a regulated process, is often dysregulated in cancer, leading to uncontrolled cell proliferation; therapies that activate apoptotic pathways or inhibit anti-apoptotic proteins hold promise for inducing cancer cell death [72]. Ferroptosis, on the other hand, is a distinct form of programmed cell death that is characterized by the accumulation of lipid hydroperoxides to lethal levels, leading to membrane damage and cell death [73]. It is characterized by iron and lipidic ROS/peroxides’ accumulation due to the Fenton reaction and via the loss of balance in ROS production and cell glutathione (GSH)-dependent antioxidants, which protect cells from lipid peroxidation [74]. Emerging evidence suggests that ferroptosis may have a tumor-suppressive function by removing cells that lack access to critical nutrients or have been compromised by infection or environmental stress [75]. Jiang et al. showed that γ-tocopherol treatment induced apoptosis in prostate cancer cells [76] (Figure 4). γ-tocotrienol was more effective than γ-tocopherol to inhibit prostate tumor growth in vivo [77]. Furthermore, according to Gopalan et al., γ-tocopherol triggers apoptosis and enhances the expression of death receptor-5 in human breast cancer cells [78]. Mechanistic analysis revealed that treatment with γ-tocopherol leads to the intracellular buildup of dihydroceramides and dihydrosphingosine in prostate cancer cells [79], as well as ceramides and dihydroceramides in breast cancer cells [78]. The use of chemical inhibitors to block de novo ceramide biosynthesis significantly reduced γ-tocopherol’s apoptotic effects [78,79]. These results highlight that γ-tocopherol’s anticancer properties are mediated through its influence on the de novo synthesis of sphingolipids. Regarding its metabolites, while γ-tocopherol -13′-COOH has not been specifically investigated, similar metabolites such as δT-13′-COOH and δTE-13′-COOH have been shown to inhibit the proliferation of colon cancer cells, while having minimal effect on normal cells [80].

Moreover, Nrf2 plays a crucial role in protecting against ferroptosis. In fact, Nrf2 upregulate genes involved in antioxidant defense, iron metabolism, and NADPH synthesis. Through this regulation, Nrf2 enhances glutathione synthesis, supporting glutathione peroxidase 4 (GPX4) activity, the primary defense against lipid peroxidation. Nrf2 also regulates iron metabolism by increasing ferroportin expression, which reduces intracellular iron levels and limits ROS generation. Furthermore, it promotes NADPH regeneration via NQO1, an essential process for maintaining GPX4 function [81]. As the downregulation of Nrf2 is generally associated with increased susceptibility to ferroptosis, the importance of maintaining Nrf2 activity becomes apparent. In this regard, it has been demonstrated that a γ-tocopherol-rich mixture of tocopherols (γ-TmT) suppresses prostate tumor development in transgenic adenocarcinoma of the mouse prostate (TRAMP) by epigenetically maintaining Nrf2 expression. Specifically, γ-TmT inhibits DNA methyltransferases (DNMT1, DNMT3A, and DNMT3B), preventing hypermethylation of the Nrf2 promoter and, thus, preserving Nrf2 activity. This sustained Nrf2 expression leads to increased levels of downstream antioxidant and detoxification enzymes, ultimately protecting against prostate tumorigenesis [43] (Figure 4).

In addition to ferroptosis, γ-tocopherols have also been observed to modulate apoptosis, which is particularly relevant in the context of cancer. Apoptosis, a tightly regulated form of programmed cell death, plays a crucial role in maintaining tissue homeostasis by eliminating damaged or unwanted cells. Evasion of apoptosis is recognized as a hallmark of cancer, enabling uncontrolled cell proliferation and tumor development [82]. Mechanistic studies have revealed that the pro-apoptotic effect of γ-tocopherol is mediated through its ability to disrupt the balance between pro- and anti-apoptotic proteins, such as Bcl-2 and Bax [83]. In this study, it has been demonstrated that the combination of γ-tocopherol and 5-fluorouracil (5-FU) modulated the expression of the apoptosis-related genes as it increased the expression of Bax, a pro-apoptotic protein, while decreasing the expression of Bcl-2, an anti-apoptotic protein [83]. This disruption leads to the release of cytochrome c from the mitochondria, triggering the activation of the intrinsic mitochondrial apoptotic pathway. In fact, combining γ-tocopherol and lovastatin in HT-29 cells led to a loss of mitochondrial membrane potential and cytochrome c release, indicating the involvement of the intrinsic apoptotic pathway. Caspase-3 activation, a key executioner of apoptosis, was also observed [84]. In breast cancer cells, it has been demonstrated that γ-tocopherol induces apoptosis by activating the JNK/CHOP/DR5 signaling pathway. In fact, γ-tocopherol treatment activates JNK, a stress-activated protein kinase critical for apoptosis. JNK upregulates CHOP (GADD153), a transcription factor induced by cellular stress, which promotes apoptosis by regulating pro-apoptotic genes. CHOP also increases DR5 expression, a TNF receptor family death receptor, triggering the extrinsic apoptotic pathway and linking intrinsic and extrinsic apoptotic cascades [78].

### 5.2. Modulation of Cell Cycle

γ-tocopherol has also been shown to modulate the expression and activity of key proteins involved in cell-cycle regulation and survival. A study by Gysin et al. revealed that γ-tocopherol induces cell-cycle arrest at the G0/G1 phase, halting progression to the S phase and subsequent division. This transition, controlled by cyclins D and E, is significantly impaired as γ-tocopherol reduces these proteins’ levels. This cycle arrest is observed in various cancer cell lines, including prostate (DU145 and LNCaP), colon, and bone cancer cells. Furthermore, γ-tocopherol inhibits DNA synthesis by 55% in DU-145 and Caco-2 cells, aligning with its impact on cell-cycle progression [85]. Another study explored the effects of α- and γ-tocopherol on the human androgen-independent PC-3 prostate cancer cell line, revealing that both forms of tocopherol inhibited cell growth, with γ-tocopherol showing significantly higher potency. Also, treatment with γ-tocopherol decreased levels of cyclins D1 and E and increased both the expression and activity of transglutaminase 2 (TG2), which is an enzyme associated with cell differentiation and apoptosis. Moreover, both tocopherols reduced the number of cells entering the S phase of the cell cycle, thereby slowing down cell division, with the effect of γ-tocopherol being more prominent [86]. Interestingly, while Sato et al. explored the combined effects of δ-tocotrienol (δ-T3) and γ-tocopherol on prostate cancer cell growth using the androgen-sensitive LNCaP cell line, they showed that γ-tocopherol-induced arrest at the G2/M phase, suggesting a possible correlation with alterations in the expression of cyclin B, a key regulator of this phase [87] (Figure 4).

### 5.3. Modulation of Cell Proliferation

In a study investigating the effect of a γ-TmT, containing 58% of γ-tocopherol, on estrogen (E2) -induced mammary hyperplasia in rats, Smolarek et al. showed that γ-TmT inhibits cell proliferation by downregulating estrogen receptor α (ERα) expression, reducing serum estrogen levels, and upregulating peroxisome proliferator-activated receptor (PPARγ) and Nrf2 [42]. Furthermore, Campbell et al. found that γ-tocopherol significantly increased both mRNA and protein expression of PPARγ in SW480 human colon cancer cells in a dose-dependent manner [88]. PPARγ is a ligand-activated transcription factor that regulates gene expression by binding to DNA. It plays a crucial role in carcinogenesis due to its involvement in the normal growth and differentiation of cells. Researchers believe that an agent’s ability to modulate PPARγ could be significant, potentially leading to clinical implications for cancer chemoprevention, particularly in the colon [88]. When compared to α-tocopherol in vitro, γ-tocopherol showed a higher potency in inhibiting the growth and proliferation of MCF-7 breast cancer cells. Furthermore, γ-tocopherol significantly decreased ERα protein levels in these cells, which is a key finding since estrogen, acting through ERα, promotes the growth of many breast cancers. Indeed, by lowering ERα levels, γ-tocopherol disrupts this growth-promoting pathway. In vivo, when included in the diet of mice bearing MCF-7 xenografts, γ-tocopherol also markedly suppresses tumor growth [83] (Figure 4).

## 6. Clinical Trials and Therapeutic Potential

γ-tocopherol has been investigated in numerous clinical trials for its potential therapeutic benefits, particularly in managing inflammatory and oxidative stress-related conditions. Trials have demonstrated promising results, including increased serum levels of γ-tocopherol and its metabolite γ-CEHC, reduced nitrosative stress, and inhibition of NF-κB and JNK signaling pathways. γ-tocopherol supplementation also decreased pro-inflammatory cytokines such as IL-1β, IL-6, IL-8, and TNF-α, as well as neutrophil and eosinophil counts elevated by environmental exposures. Moreover, it prevented wood-smoke–induced airway inflammation and improved mucociliary clearance while reducing total sputum mucins. Additionally, γ-tocopherol showed vascular benefits, such as improved flow-mediated dilation, although some trials reported no significant changes in nitric oxide pathways or other oxidative stress markers. These findings highlight its potential role as an anti-inflammatory and antioxidant therapeutic agent. (Figure 5, Table 2).

## 7. Conclusions

γ-tocopherol, a significant isoform of vitamin E, emerges as a potent antioxidant with diverse protective effects against oxidative stress-induced damage. Its unique structural properties, including the chromanol ring and hydrophobic phytyl tail, enable its integration into lipid membranes, where it plays a pivotal role in neutralizing ROS and RNS. Furthermore, the anti-inflammatory activity of γ-tocopherol is also closely intertwined with its antioxidant properties. By neutralizing reactive oxygen and nitrogen species, γ-tocopherol not only reduces oxidative stress, a key driver of inflammation, but also modulates signaling pathways, such as NF-κB and Nrf2, thereby suppressing the production of pro-inflammatory cytokines and mediators. These combined effects underscore the critical role of its antioxidant capacity in contributing to its anti-inflammatory efficacy. Additionally, γ-tocopherol demonstrates remarkable efficacy in reducing lipid peroxidation markers, such as MDA and 8-isoprostane, across various biological systems, underscoring its potential in preventing cellular and tissue damage. Preclinical studies highlight its protective effects against DNA oxidative damage by reducing markers like 8-oxo-dG and γH2AX in multiple models, indicating its role in safeguarding genetic material. Nevertheless, further investigation into the precise molecular mechanisms and their influence on DNA structure and stability would be valuable. Additionally, γ-tocopherol’s antioxidant effects extend to enhancing the body’s defense systems, potentially offering therapeutic benefits for conditions linked to oxidative stress, including metabolic syndrome, environmental toxin exposure, and chronic inflammation.

Despite these promising findings, the complexity of its mechanisms and interactions with other antioxidants, such as α-tocopherol, necessitates further research. Future studies should focus on elucidating its precise molecular pathways and exploring its clinical applicability as a dietary supplement or therapeutic agent. Overall, γ-tocopherol stands out as a vital natural compound with immense potential to combat oxidative stress and related disorders, reinforcing the importance of diet-derived antioxidants in health promotion and disease prevention. Given the promising antioxidant and protective effects of γ-tocopherol demonstrated in various studies, the exploration of nanoformulation strategies becomes essential. These approaches could enhance its stability, bioavailability, and targeted delivery, addressing the limitations of conventional formulations. Since the existing research on nanoformulated γ-tocopherol is limited but encouraging, further investigations are warranted to optimize its therapeutic potential and broaden its application in oxidative stress-related disorders.

## Figures and Tables

**Figure 1 molecules-30-00653-f001:**
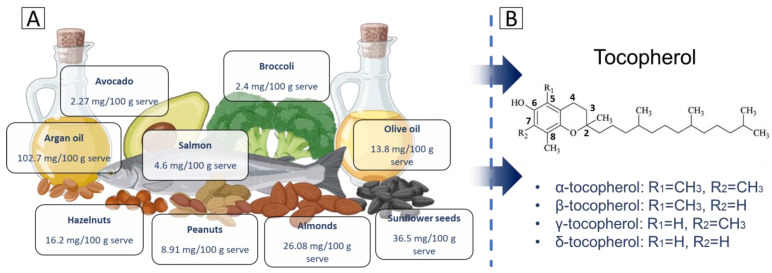
Dietary sources and structural variants of tocopherols: (**A**) key dietary sources of tocopherols, including argan oil, avocado, broccoli, salmon, hazelnuts, almonds, sunflower seeds, peanuts, and olive oil, with their respective tocopherol content per serving. (**B**) The chemical structure of tocopherol, highlighting the differences in methyl group positions that distinguish alpha (α), beta (β), gamma (γ), and delta (δ) tocopherols.

**Figure 2 molecules-30-00653-f002:**
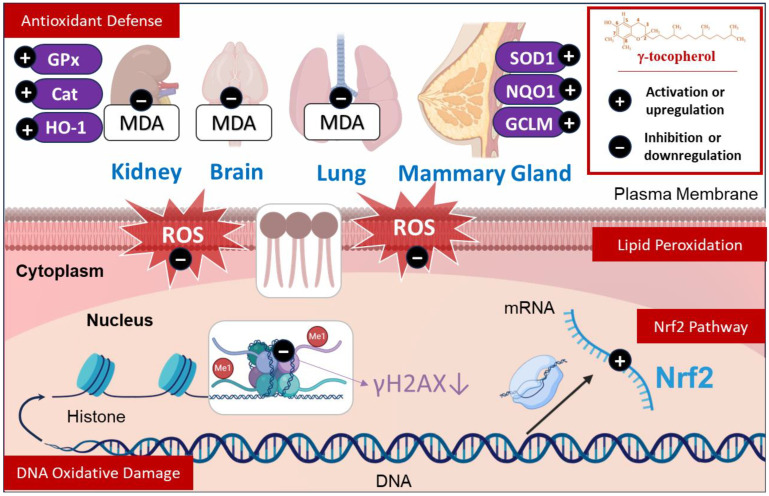
γ-tocopherol’s therapeutic effects in oxidative stress: γ-tocopherol’s role in reducing lipid peroxidation (via suppression of MDA and ROS production), mitigating DNA oxidative damage (indicated by reduced γH2AX, 8-oxo-dG, and PHIP-induced damage), enhancing antioxidant defense systems (elevating levels of GPx, SOD1, Cat, NQO1, HO-1, and GCLM), and its involvement in the Nrf2 pathway (inducing Nrf2 activation and antioxidant gene expression).

**Figure 3 molecules-30-00653-f003:**
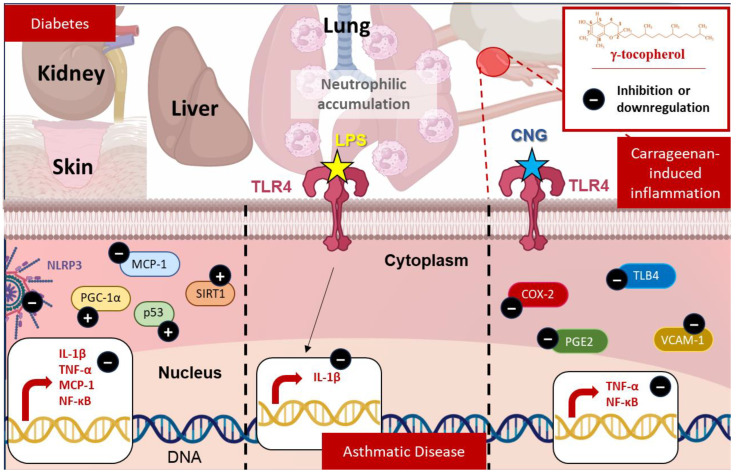
γ-tocopherol’s therapeutic effects in inflammatory and metabolic diseases: γ-tocopherol modulates inflammation and oxidative stress in diabetes (by reducing CRP, MCP-1, IL-1β, TNF-α, and NLRP3, while enhancing p53, PGC-1α, and SIRT1), carrageenan-induced inflammation (by lowering LTB4, TNF-α, PGE2, COX-2, VCAM-1, and NF-κB), and asthmatic disease (by decreasing neutrophilic accumulation and IL-1β levels in LPS-induced inflammation).

**Figure 4 molecules-30-00653-f004:**
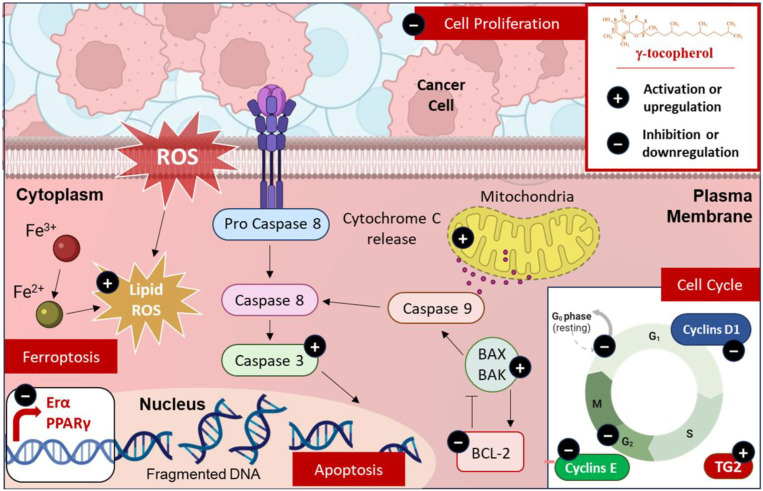
γ-tocopherol’s therapeutic effects in cancer: γ-tocopherol promotes cell death through apoptosis and ferroptosis, evidenced by interactions of Bcl-2, Bax proteins, and activated Caspase 3. It also highlights its impact on the cell cycle, notably inhibiting progression via cyclins D1 and E, which affects the resting (G0) phase and other stages. Lastly, γ-tocopherol influences cell proliferation by downregulating estrogen receptor α (Erα) and PPARγ, thereby reducing cancer cell growth in experimental models.

**Figure 5 molecules-30-00653-f005:**
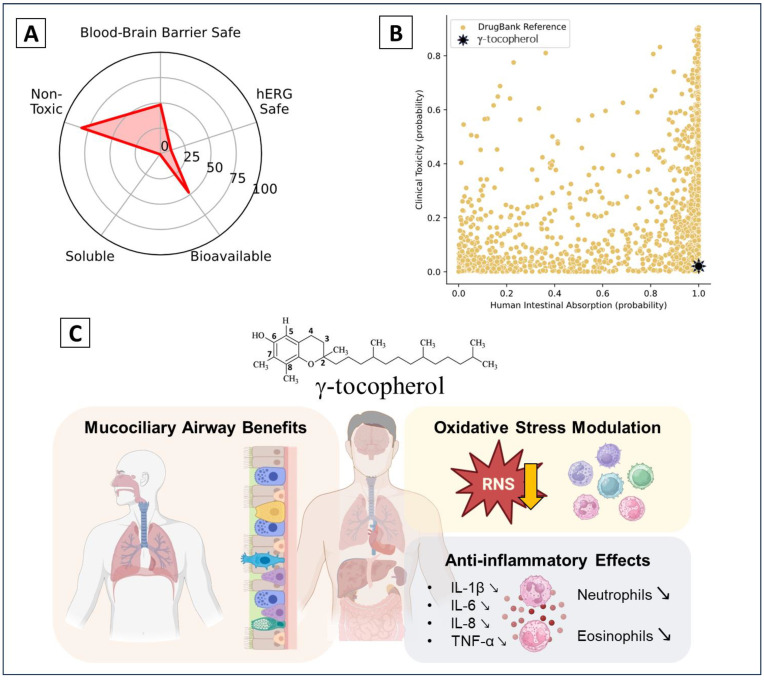
γ-tocopherol’s properties and effects. (**A**) Favorable pharmacokinetic and safety profile (bioavailability, solubility, and non-toxicity)—information obtained from https://admet.ai.greenstonebio.com (accessed on 16 December 2024); (**B**) high absorption and low toxicity compared to other compounds—information obtained from https://admet.ai.greenstonebio.com (accessed on 16 December 2024); and (**C**) clinical trial’s key biological effects.

**Table 1 molecules-30-00653-t001:** Antioxidant actions and mechanisms across different animal models.

Model	Antioxidant Action	Mechanisms	References
Human (Plasma) Mice (Kidney, Liver) Rat (Plasma) Sheep (Lung) Chicken (Plasma, Liver)	Anti-lipid peroxidation	↓ MDA ↓ MDA+HNE ↓ Lipid peroxide ↓ 8-isoprostane	[32,33,34,35,36]
Mice (Colon tumor and tissue adjacent, Colon) Rat (Mammary gland)	Anti-DNA damage	↓ 8-oxo-dG ↓ γH2AX	[35,37,38,39]
Mice (Kidney, Liver, Prostate, mammary tumor) Rat (Plasma, Mammary gland, aorta) PBM Cells	Defense antioxidant	↑ mRNA SOD1 ↑ SOD activity ↓ Superoxide generation ↓ ROS ↔/↑ Catalase ↔/↑ GPx ↔ HO-1 ↑ NQO1 ↑ GCLM ↑ HO-1 ↔/↑ Nrf2 Inhibition CpG hypermethylation in Nrf2 promoter	[32,35,40,41,42,43]

MDA: malondialdehyde, MDA+HNE: malondialdehyde + 4-hydroxy-2-nonenal, lipid peroxide: lipid peroxidation products, 8-isoprostane: 8-iso-prostaglandin F2α, 8-oxo-dG: 8-Oxo-2′-deoxyguanosine, γH2AX: phosphorylated H2A.X histone variant, mRNA SOD1: messenger RNA for superoxide dismutase 1, SOD activity: superoxide dismutase activity, superoxide generation: generation of superoxide anions (O_2_•−), ROS: reactive oxygen species, catalase: catalase enzyme activity, GPx: glutathione peroxidase, HO-1: heme oxygenase-1, NQO1: NAD(P)H quinone dehydrogenase 1, GCLM: glutamate-cysteine ligase modifier subunit, Nrf2: nuclear factor erythroid 2-related factor 2, and PBM cells: peripheral blood mononuclear cells; ↑: increase, ↓: decrease, and ↔: standardization.

**Table 2 molecules-30-00653-t002:** Clinical trials using γ-tocopherol found on https://clinicaltrials.gov (accessed on 25 December 2024).

Intervention/Treatment	Phase	Actual Enrollment	Identifier	Responsible Party	Results
3 doses of 1400 mg γ-tocopherol (2 capsules, each is 700 mg), at 12 h intervals	Not Applicable	10 subjects 18 to 50 years	NCT02610829	University of North Carolina, Chapel Hill	Increased serum levels of γ-tocopherol and its metabolite γ-CEHC; Reduced levels of nitrosative stress; Inhibit NF-κB and JNK pathways; Reduced cytokine secretion in PBMCs; Reduced reactive nitrogen oxide species (RNOS) generation.
Oral doses 1200 mg of γ-tocopherol (2 capsules of the γ-tocopherol enriched vitamin E preparation)	Phase 1	25 subjects 18 to 50 years	NCT00836368	Chapel Hill, North Carolina, United States	No Results Posted
14 days of daily high dose (1200 mg) γ-tocopherol. Subjects will receive a 14 days supply (28 softgel capsules, approximately 600 g of γ-tocopherol each	Phase 1	8 subjects 18 to 50 years	NCT00466596	University of North Carolina, Chapel Hill	No Results Posted
γ-tocopherol 1400 mg, taken as 2700 mg capsules every 12 h for a total of 4 doses	Phase 2	18 subjects 18 to 45 years	NCT02911688	University of North Carolina, Chapel Hill	Reduction in neutrophil and eosinophil counts elevated by ozone exposure; Decreased levels of pro-inflammatory cytokines IL-1β, IL-6, IL-8, and TNF-α induced by ozone.
γ-tocopherol Maxi Gamma softgels 1200 mg	Phase 1	18 subjects 18 to 50 years	NCT00631085	University of North Carolina, Chapel Hill	Decreased LPS-induced neutrophil and eosinophil percentage; Decreased IL-1β levels.
1400 mg of γ-tocopherol -enriched supplement once daily for 7 days	Phase 2	16 subjects 18 to 45 years	NCT03444298	University of North Carolina, Chapel Hill	Prevention of wood smoke–induced eosinophilic airway inflammation.
600 mg and 1200 mg of γ-tocopherol safety	Phase 1	16 subjects 18 to 50 years	NCT00386178	University of North Carolina, Chapel Hill	No Results Posted
1200 mg of γ-tocopherol daily for 14 days	Phase 1 Phase 2	23 subjects 18 to 50 years	NCT02104505	University of North Carolina, Chapel Hill	γ-tocopherol treatment prevented the slowdown in mucociliary clearance observed 4 h post-challenge in the placebo group; Total sputum mucins (excluding MUC5AC) were reduced 24 h post-challenge with γ-tocopherol treatment compared to placebo.
One administration of 500 mg γ-tocopherol	Phase 1 Phase 2	67 subjects 18 to 60 years	NCT01314443	University of Connecticut	Improved flow-mediated dilation; No changes in nitric oxide (NO•)-mediated pathways or other oxidative stress markers (e.g., oxLDL, MDA).

## Data Availability

Not applicable.

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
