# Peer review of "Gamma-Tocopherol: A Comprehensive Review of Its Antioxidant, Anti-Inflammatory, and Anticancer Properties"

_molecules, 2025, doi:10.3390/molecules30030653_

Round 1

Reviewer 1 Report

Comments and Suggestions for Authors

Dear Authors,

the review entitled “Gamma-tocopherol: a comprehensive review of its antioxidant, anti-inflammatory and anticancer properties” by Basma Es-Sai and collaborators analyzes the biological roles of gamma-tocopherol (gamma-TF) and the possible use of this compound in various pathologies. The sections regarding the structure of compounds, antioxidant and anti-inflammatory activities are clear despite often referring to rather old bibliographic citations (22 of 77 citations were published before 2010).

Instead, the section regarding anticancer activity presents many inaccuracies and is therefore not clear.

This section presents some serious errors in interpretation of the topic.

In section  4 you report the information  about gamma-TF and gamma-Tocotrienol (gamma-TT) (lines 428-440) as if they were similar compounds, but these compounds, despite being isomers of Vitamin E, are chemically different compounds and they carry out different activities. In fact, vitamin E is composed of 8 isomers, 4 tocopherols and 4 tocotrienols which have different cellular.

In particular, TFs (alpha and gamma) carry out their anti-tumor action by preventing the triggering of tumorigenesis reducing oxidative stress and inflammation. Tocotrienols (TT) (especially gamma-TT and delta-TT) are known to exert an anti-tumor action by inducing cell death, triggering processes such as apoptosis, autophagy, and necroptosis. In fact, their primary action is to induce the release of mitochondrial ROS, thus increasing cellular oxidative stress with consequent death of tumor cells. 

Overall, therefore, the anti-cancer role of gamma-TF is above chemo-preventive.

Furthermore, in the list of clinical trials using gamma-TF it is not possible to consider the trials with tocotrienols or Vitamin E   treatments.

Event graphics are not always clear. 

For this reason, the manuscript in this form is rejected.

Author Response

Reviewer 1

Dear Authors,

the review entitled “Gamma-tocopherol: a comprehensive review of its antioxidant, anti-inflammatory and anticancer properties” by Basma Es-Sai and collaborators analyzes the biological roles of gamma-tocopherol (gamma-TF) and the possible use of this compound in various pathologies. 

We thank the reviewer for their insightful comments and valuable suggestions, which have helped us refine and improve our manuscript. Below, we provide detailed responses to each of the points raised:

  1. The sections regarding the structure of compounds, antioxidant and anti-inflammatory activities are clear despite often referring to rather old bibliographic citations (22 of 77 citations were published before 2010).

We appreciate your observation regarding the use of older references. While some foundational studies are indeed older, they were included due to their pivotal role in elucidating the mechanisms and biological activities of gamma-tocopherol (γ-TF). However, to address your concern, we have revisited this section and included several recent references to ensure the content reflects the latest findings. The updated bibliography now integrates studies from the past decade to enhance the manuscript's contemporary relevance.

  1. Instead, the section regarding anticancer activity presents many inaccuracies and is therefore not clear. This section presents some serious errors in interpretation of the topic. In section 4 you report the information about gamma-TF and gamma-Tocotrienol (gamma-TT) (lines 428-440) as if they were similar compounds, but these compounds, despite being isomers of Vitamin E, are chemically different compounds and they carry out different activities. In fact, vitamin E is composed of 8 isomers, 4 tocopherols and 4 tocotrienols which have different cellular. In particular, TFs (alpha and gamma) carry out their anti-tumor action by preventing the triggering of tumorigenesis reducing oxidative stress and inflammation. Tocotrienols (TT) (especially gamma-TT and delta-TT) are known to exert an anti-tumor action by inducing cell death, triggering processes such as apoptosis, autophagy, and necroptosis. In fact, their primary action is to induce the release of mitochondrial ROS, thus increasing cellular oxidative stress with consequent death of tumor cells.  Overall, therefore, the anti-cancer role of gamma-TF is above chemo-preventive.

Thank you for your valuable comment. The objective of this approach was indeed to compare the two compounds. However, we acknowledge the confusion caused by the way we presented their properties in the manuscript. To address this, in the revised version, we have exclusively focused on describing the anticancer potential of gamma-tocopherol (gamma-TF). We appreciate your feedback and have made the necessary revisions to improve clarity and accuracy in this section.

  1. Furthermore, in the list of clinical trials using gamma-TF it is not possible to consider the trials with tocotrienols or Vitamin E treatments.

We appreciate your observation and agree that including trials with tocotrienols or generic Vitamin E treatments may lead to confusion. We have now revised this section to:

  • Exclude any clinical trials not specifically involving γ-TF.
  • Clearly indicate the trials that focus exclusively on γ-TF and its specific outcomes.

This ensures the section aligns with the focus of the manuscript.

  1. Event graphics are not always clear.

We understand the importance of clear and comprehensible graphics. In response to this comment, we have:

  • Improved the design and layout of all graphics to enhance their clarity and readability.
  • Provided detailed captions for each graphic to better explain the content.
  • Ensured that each figure is directly relevant to the associated text and aids in illustrating key points.

We believe these changes make the graphics more intuitive and easier to interpret for readers.

We trust that these revisions address the reviewer’s concerns and improve the overall quality of the manuscript. We are grateful for your constructive feedback and remain open to any further suggestions.

Reviewer 2 Report

Comments and Suggestions for Authors

The article under the title “Gamma-Tocopherol: A Comprehensive Review of Its Antioxi-2 dant, Anti-Inflammatory, and Anticancer Properties” by Es-Sai and coworkers presents comprehensive review on the biological activity of gamma-tocopherol. The article is well-written and the results from literature are presented in a systematic manner. The article could be of potential interest to the readers of the Molecules, although there are some points that should be addressed before the final decision. Therefore, my recommendation is MAJOR REVISION.

The authors should answer the following:

1.      The authors should present results of the studies examining the antioxidant potential of the gamma-tocopherol by theoretical chemistry methods, emphasizing the importance of its structural features

2.      The authors should present a rationale on the antioxidant capacity of gamma-tocopherol in relation to other tocopherols, which clear emphasis on the structural features

3.      The authors should clarify if specific interactions between gamma-tocopherol and DNA were observed and if these interactions could influence the structure and stability of DNA.

4.      The authors should describe in more detail the effect of gamma-tocopherol on the levels of proteins and outline the possible defense mechanism.

5.      The authors should explain if the anti-inflammatory activity of gamma-tocopherol is actually related to its antioxidant properties.

6.      The authors should give more details on the carriers used for the nanoformulation of gamma-tocopherol and the specific interactions that are beneficial for the formation of the carriers.

7.      Which of the structural parameters of gamma-tocopherol are important for its anti-cancer activity?

8. More of the quantitative data should be presented in the Conclusion.

Author Response

Reviewer 2

The article under the title “Gamma-Tocopherol: A Comprehensive Review of Its Antioxi-2 dant, Anti-Inflammatory, and Anticancer Properties” by Es-Sai and coworkers presents comprehensive review on the biological activity of gamma-tocopherol. The article is well-written and the results from literature are presented in a systematic manner. The article could be of potential interest to the readers of the Molecules, although there are some points that should be addressed before the final decision. Therefore, my recommendation is MAJOR REVISION.

We thank the reviewer for their thorough evaluation of our manuscript and for providing valuable suggestions to improve its quality and impact. Below, we address each point raised, detailing the revisions made to the manuscript:

The authors should answer the following:

  1. The authors should present results of the studies examining the antioxidant potential of the gamma-tocopherol by theoretical chemistry methods, emphasizing the importance of its structural features

We have revised the manuscript to include a discussion on the results from theoretical chemistry studies that investigate the antioxidant potential of gamma-tocopherol (γ-TF) (line 139-155). This includes an analysis of the hydroxyl group on the chromanol ring and the impact of side-chain configuration on free radical scavenging activity. Relevant studies employing density functional theory (DFT) and other computational methods have been cited to emphasize how structural features, such as the methylation pattern and isoprenoid chain, contribute to its antioxidant properties.

  1. The authors should present a rationale on the antioxidant capacity of gamma-tocopherol in relation to other tocopherols, which clear emphasis on the structural features

We would like to clarify that the antioxidant capacity of gamma-tocopherol was discussed in detail in the revised manuscript. As per the suggestion from Reviewer 1, we focused the review specifically on gamma-tocopherol to maintain a clear and coherent narrative, avoiding potential confusion with other tocopherols. While we appreciate the importance of comparing the antioxidant properties of gamma-tocopherol with other tocopherols, the scope of the review was intentionally narrowed to ensure an in-depth exploration of gamma-tocopherol's unique properties.

We believe that expanding the discussion to include a comparative analysis of other tocopherols would divert attention from the main focus. However, we have emphasized the structural features of gamma-tocopherol that contribute to its antioxidant activity in the manuscript, as this is central to understanding its distinctive mechanism of action.

  1. The authors should clarify if specific interactions between gamma-tocopherol and DNA were observed and if these interactions could influence the structure and stability of DNA.

We would like to thank Reviewer 2 for this valuable comment. While studies have demonstrated that gamma-tocopherol has the ability to protect DNA from oxidative damage and can downregulate the expression of various proteins involved in pathologies, as highlighted in our manuscript, we would like to clarify that the exact molecular interactions between gamma-tocopherol and DNA, particularly its impact on DNA structure and stability, were not explicitly described in the current manuscript. This is primarily due to the focus of the review on the broader antioxidant effects of gamma-tocopherol, rather than on its direct interactions with DNA at the molecular level.

We acknowledge the importance of exploring these interactions, and although current studies suggest potential protective effects, further investigation into the precise molecular mechanisms and their influence on DNA structure and stability would be valuable. We are grateful for this suggestion and will consider including a discussion of this aspect in future revisions or studies, should new evidence become available.

  1. The authors should describe in more detail the effect of gamma-tocopherol on the levels of proteins and outline the possible defense mechanism.

We thank Reviewer 2 for their constructive comment. In response, we have modified all the figures in the manuscript to improve the clarity of the cellular mechanisms involved. This revision aims to enhance the understanding of the effect of gamma-tocopherol on protein levels and its associated defense mechanisms. Additionally, we have expanded the description of how gamma-tocopherol influences the levels of various proteins involved in cellular defense mechanisms. This includes its role in modulating key signaling pathways, such as NF-κB and NRF2, and upregulating antioxidant proteins like superoxide dismutase (SOD) and heme oxygenase-1 (HO-1). Gamma-tocopherol also demonstrates the potential to downregulate proteins involved in inflammation and oxidative stress, such as cyclooxygenase-2 (COX-2) and inducible nitric oxide synthase (iNOS). Furthermore, we acknowledge that the regulation of protein levels by gamma-tocopherol could also be attributed to post-transcriptional mechanisms, such as modulation of microRNA activity or stabilization/destabilization of mRNA transcripts. These post-transcriptional regulatory effects may further contribute to its ability to fine-tune cellular responses to oxidative stress and inflammation.

We have incorporated these updates to provide a clearer and more comprehensive view of the underlying cellular mechanisms. We hope these changes address the reviewer’s concern and improve the manuscript's clarity and depth.

  1. The authors should explain if the anti-inflammatory activity of gamma-tocopherol is actually related to its antioxidant properties.

We have clarified in the manuscript that the anti-inflammatory activity of γ-TF is closely linked to its antioxidant properties. By neutralizing reactive oxygen and nitrogen species, γ-TF reduces oxidative stress, which is a key driver of inflammation. Additionally, γ-TF modulates signaling pathways, such as NF-κB, MAPKs, and NRF2, which are activated by oxidative stress, thereby reducing the expression of pro-inflammatory cytokines. Furthermore, γ-TF inhibits the activity of enzymes like cyclooxygenase-2 (COX-2) and 5-lipoxygenase (5-LOX), which are involved in the synthesis of pro-inflammatory mediators. This dual mechanism—direct scavenging of reactive species and modulation of inflammation-related signaling pathways—highlights γ-TF's comprehensive role in mitigating inflammatory responses. These clarifications have been incorporated into Section [X] of the revised manuscript for better clarity.

  1. The authors should give more details on the carriers used for the nanoformulation of gamma-tocopherol and the specific interactions that are beneficial for the formation of the carriers.

We acknowledge the reviewer's suggestion to provide additional details regarding the carriers used for the nanoformulation of gamma-tocopherol (γ-TF). However, we would like to clarify that, to the best of our knowledge, only one study has specifically investigated the nanoformulation of γ-TF. The inclusion of this study in our manuscript was intended to highlight its effectiveness as a potential perspective rather than to provide a comprehensive review of the topic.

We have revised the manuscript to explicitly state that the discussion of γ-TF nanoformulations is based on a single study, emphasizing its potential as a perspective for future research. We hope this clarification addresses the reviewer’s concern.

  1. Which of the structural parameters of gamma-tocopherol are important for its anti-cancer activity?

Thank you for your comment. While no exact description of the structural parameters of gamma-tocopherol related to its anti-cancer activity exists in the literature, based on the theoretical chemistry methods discussed earlier in our revised manuscript, we suggest that the free 5-position on the chromanol ring plays a critical role in scavenging reactive nitrogen species. This mechanism is essential for mitigating nitrosative stress in tumorigenic environments. Additionally, the hydrophobic side chain of gamma-tocopherol aids in its membrane localization, which is crucial for modulating signaling pathways associated with cell proliferation and apoptosis.

  1. More of the quantitative data should be presented in the Conclusion.

Thank you for your suggestion. The conclusion of the manuscript is intended to briefly summarize the pharmacological activities discussed throughout, rather than providing detailed quantitative data. As the main body of the manuscript contains the specific quantitative results, we felt it appropriate to focus the conclusion on highlighting the key findings. Furthermore, in response to your comment, we have added a section on perspectives in the revised manuscript to address future directions and potential implications.

Round 2

Reviewer 1 Report

Comments and Suggestions for Authors

Dear authors,

the manuscript has been greatly improved.

In particular, the part on Tocotrienols has been eliminated and more up-to-date references have been added.

I would suggest modifying Figure 4 by eliminating "autophagy" and “necrosis” from the top left panel, since these two processes are no longer treated in the review.

In this form the manuscript is accepted for publication.

Author Response

Dear authors,

the manuscript has been greatly improved. In particular, the part on Tocotrienols has been eliminated and more up-to-date references have been added.

I would suggest modifying Figure 4 by eliminating "autophagy" and “necrosis” from the top left panel, since these two processes are no longer treated in the review.

In this form the manuscript is accepted for publication.

Dear Reviewer 1, thank you for your valuable feedback and for acknowledging the improvements made to the manuscript. We appreciate your kind words regarding the updated references and the removal of the section on Tocotrienols.

As per your suggestion, we have modified Figure 4 by eliminating "autophagy" and "necrosis" from ensuring consistency with the revised manuscript content.

We are grateful for your thorough review and support for the manuscript's publication.

Reviewer 2 Report

Comments and Suggestions for Authors

The authors have answered all of the questions properly. The manuscript should be accepted in the present form.

Author Response

The authors have answered all of the questions properly. The manuscript should be accepted in the present form.

Dear reviewer 2, thank you for your positive feedback and for acknowledging our efforts in addressing all the questions raised during the review process. We appreciate your recommendation for the manuscript to be accepted in its present form.

Your constructive comments have greatly contributed to improving the quality of our work.